# Impact of catch-up human papillomavirus vaccination on cervical conization rate in a real-life population in France

**Antoine Eliès**[1]*, **Claire Bonneau**[1,2], **Sophie Houzard**[1], **Roman Rouzier**[1,2], **Delphine Héquet**[1,2]

**1** Department of Surgical Oncology, Institut Curie, Saint-Cloud, France, **2** Université Versailles St-Quentin, Université Paris-Saclay, Montigny-le-Bretonneux, France

* antoineelies@free.fr

## Abstract

### Objective

To evaluate the impact of catch-up human papillomavirus (HPV) vaccination on conization rates in France in a large population-based study.

### Methods

We conducted a retrospective real-life cohort study on data collected prospectively by French National Health Insurance. Echantillon généralistes des bénéficiaires (EGB) is a database composed of demographic and health care utilization data for a 1/97th sample of the French population. We extracted data about all women born between 1983 and 1991, corresponding to the catch-up population (vaccination after 14 years old) at the time of implementation of HPV vaccination. The primary outcome was the occurrence of conization (all types of procedures) compared between vaccinated and non-vaccinated women.

### Results

The cohort consisted of 42,452 women. Vaccination coverage (at least one dose) was low (9.8%, n = 4,129), but increased with time from vaccine implementation, from 0% in the 1983 cohort to 31% in the 1991 cohort. The conization rate was 1% for the overall population. The risk of conization for women between the ages of 19 and 30 years was reduced in the vaccinated group with a Hazard Ratio (HR) of 0.59 (95% CI[0.39–0.90]; p = 0.043).

### Conclusions

With a 10-year follow-up, catch-up HPV vaccination is associated with risk reduction of conization between the ages of 19 and 30.

**Data Availability Statement:** This study involved third-party data (National Health Insurance – EGB). Therefore, we do not have the rights to share these third-party data, in accordance with regulations

SNDS (Système National des Données de Santé). (Law governing the dissemination of SNDS data : L. 1461-1 III du code de la santé publique R. 1461-1 du code de la santé publique For more information: https://www.snds.gouv.fr/SNDS/Finalites-autorisees). To access the EGB data, please visit this website to apply for dataset : https://documentation-snds.health-data-hub.fr/introduction/03-acces-snds.html#les-acces-permanents. The authors had no special access privileges to the data.

**Funding:** The authors received no specific funding for this work.

**Competing interests:** The authors have declared that no competing interests exist.

## Introduction

Cervical cancer remains a public health issue in France, as it affects young women with a peak incidence at the age of 42 years [1, 2]. The severity of cervical cancer is related to the high morbidity associated with treatment, including chemotherapy, radiation therapy and surgery. Conization is part of the treatment of cervical dysplasia and early stage cancer. In women who have not always completed their pregnancy plans, this procedure is associated with obstetrical morbidity.

Almost all cervical cancers are induced by human papillomavirus (HPV). The genotypes most frequently involved are genotypes 16 and 18, responsible for about 70% of all cervical cancers. Sexual transmission mainly occurs during the first year of sexual activity. The natural history of HPV infection consists of slow progression either to spontaneous cure or possible dysplasia and subsequent cervical cancer.

HPV vaccination directed against the genotypes responsible for cervical cancer has been available since 2006 and has been reimbursed since 2007 in France. Its effectiveness is based on extensive vaccination coverage providing individual protection and herd immunity [3, 4]. However, vaccination coverage remains low in France, as less than 30% of 16-year-old girls born between 1996 and 2000 received a complete immunization plan in 2017 [5]. This poor take-up could be explained by a different approach to vaccination compared to that adopted in other countries (lack of a school-based vaccination program), as well as mistrust of health professionals and the general public towards the safety and effectiveness of the vaccine [6–8]. However, several studies have demonstrated the effectiveness of vaccination on genital warts [9–14] and especially on cervical dysplasia in Australia [15, 16], Canada [17], Sweden [18] and Denmark [19]. More recently, two large cohort studies showed a reduced risk of invasive cervical cancer among girls and women vaccinated in Sweden [20] and in Denmark [21]. However, no such studies have yet been conducted in France. The objective of the present study was to evaluate the impact of HPV vaccination in a real-life population.

## Materials and methods

We conducted a retrospective real-life cohort study, i.e. on a large sample of the French population, on prospectively collected data.

### Data sources and study period

Data were extracted from the French National Health Insurance database called *Echantillon Généraliste des Bénéficiaires* (EGB), which is a permanent sample of the population covered by national health insurance. French health care insurance is a universal health care meaning all residents are assured access to health care. Our institution has permanent access to the EGB given by its governance (ministerial steering), for the purposes of conducting studies on anonymous data. This study was reviewed and approved by an ethics committee: Curie Institutional Research Data Committee. The EGB is a random sample representing 1/97th of the population. Individuals are randomly selected on the social security number, the distribution by age group and sex is equivalent to the national distribution. The EGB includes people covered by the general social security system, i.e. all workers and unemployed persons (excluding farmers), this population representing more than 92% of the French population. This database includes anonymous sociodemographic characteristics, all reimbursed medical expenditure, consultations, drug prescriptions, laboratory tests and surgical procedures [22].

Data were extracted from EGB from 1st January 2006 (release of the first HPV vaccine—Gardasil® onto the market) to 31st December 2016.

We ensured that there were no published data on the efficacy of HPV vaccination in France. The following HPV terms were searched on Pubmed: "france"[All Fields] AND ("Papillomavirus Vaccines"[MeSH Terms] OR ("hpv"[All Fields] AND ("vaccin"[All Fields] OR "vaccination"[All Fields])) AND ("cancer"[All Fields] OR "neoplasms"[MeSH Terms] OR "neoplasms"[All Fields] OR "dysplasia"[All Fields]). We did not find any study similar to this research.

## Vaccination recommendations and screening procedure

During the study period, from January 2006 to December 2016, French guidelines recommended, as catch-up vaccination, an administration of a 3-dose vaccine regimen to non-sexually active 15- to 23-year-old girls or, at the latest, during the year after first sexual intercourse [23]. Vaccination is reimbursed in this indication. Recommendations for younger girls consisted of a 3-dose vaccine regimen for 14-year-old girls and, since 2014, a 2-dose regimen for 11- to 15-year-old girls [24]. Gardasil[®] and Cervarix[®], released onto the market in 2006 and 2008, respectively, were the vaccines available during the study. In France, screening for cervical dysplasia is based on Pap smear every 3 years between the ages of 25 and 65 years, with the first two smears performed at an interval of one year [25]. Women eligible for catch-up vaccination at the beginning of the vaccination campaign have now reached screening age.

## Management of dysplasia in France

The 2002 recommendations are that conization is usually indicated for Cervical Intraepithelial Neoplasia 2+ (CIN), with laser destruction or cryotherapy as an option in certain conditions (small lesions, exclusively exocervical, fully visible at colposcopy) [26]. In 2016 the recommendations were similar, specifying that conization should be a LEEP, with destructive treatments as an option (with the same criteria) [27]. In order to verify the application of these guidelines in real life, we ensured that high-grade dysplasia was treated by conization. Thus, we queried the national database on all hospital stays (outpatient or conventional). These data are freely available online (https://www.scansante.fr/). The diagnosis of severe cervical dysplasia (N872) was cross-referenced with acts of conization, destruction of cervical lesions, hysterectomies. The results are detailed in Table 1. The vast majority of severe dysplasia was treated by conization (87%), while destructive methods and hysterectomies were much less common (8% and 5% respectively).

## Study population

As girls of the HPV vaccination target population (14-year-olds) had just reached the screening age in France (14-year-old girls in 2006 were 24 years old at the end of the study period in 2016), our evaluation focused on catch-up vaccination population. The study population was therefore defined by the following criteria (Fig 1):

**Table 1. Mean number and % of annual procedure for high grade cervical dysplasia in France between 2006 and 2016.**

|  | n | % |
|---|---|---|
| Conization | 7,061 | 86.9 |
| Destruction of cervical lesions except conization | 600 | 7.6 |
| Hysterectomy | 438 | 5.4 |
| Total | 8,100 | 100 |

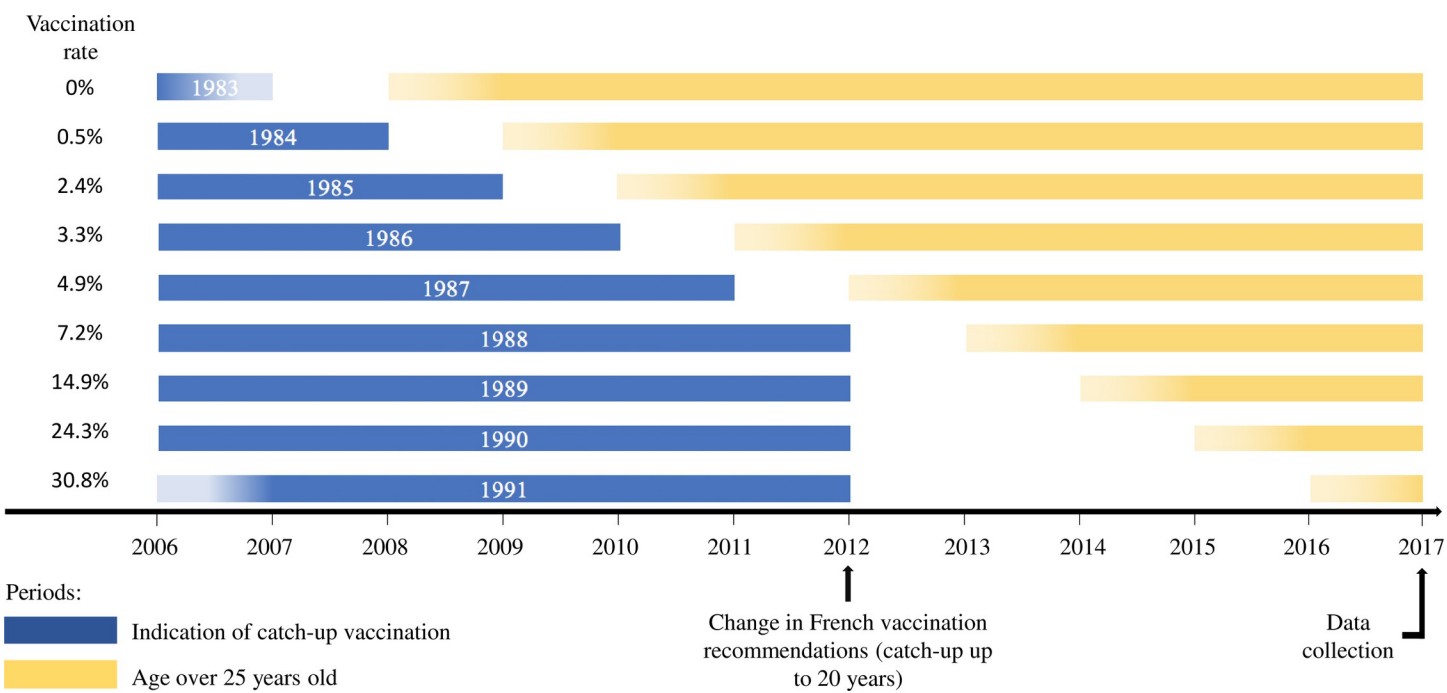

**Fig 1. Scheme of the study population according to birth cohort.** Each birth cohort (years written in white) is represented by a line. The catch-up vaccination indication period is shown in blue and the period corresponding to age greater than 25 years is shown in yellow. The catch-up vaccination rate for at least one dose is indicated at the beginning of the line.

- Catch-up vaccination target population for: women between the ages of 15 and 23 years between 2006 and 2016, i.e. born between 1983 and 2001.
- Cervical cancer screening target population: women who reached the age of 25 between 2007 and 2016, i.e. born between 1983 and 1991.

## Statistical analyses

We considered two groups: the vaccinated group, corresponding to women who had received at least one dose, and the non-vaccinated group. As cervical dysplasia is poorly documented in this database, our primary endpoint was therefore the conization rate, which reflects the rate of severe dysplasia. The main outcome was the comparison of conization rates between the non-vaccinated and vaccinated populations.

We calculated the conization rate in each group to assess the impact of vaccination, using Kaplan-Meier curves to study the risk of conization over equivalent periods of risk between the two groups. To avoid including patients who had already undergone conization prior to inclusion in the EGB, we determined the minimum age of conization in the overall population. As the minimum age at which conization was performed was 19 years, we excluded birth cohorts included in the EGB after the age of 19 (born before 1987). Birth cohorts from 1987 to 1991 were selected for the following analysis (Fig 2).

We studied the impact of potential explanatory factors for a difference between the two groups: age and number of Pap smears performed, number of general practitioner and gynaecologist visits (all reasons for consultation), and proportion of women with at least one smear at any age and after the age of 25 years old.

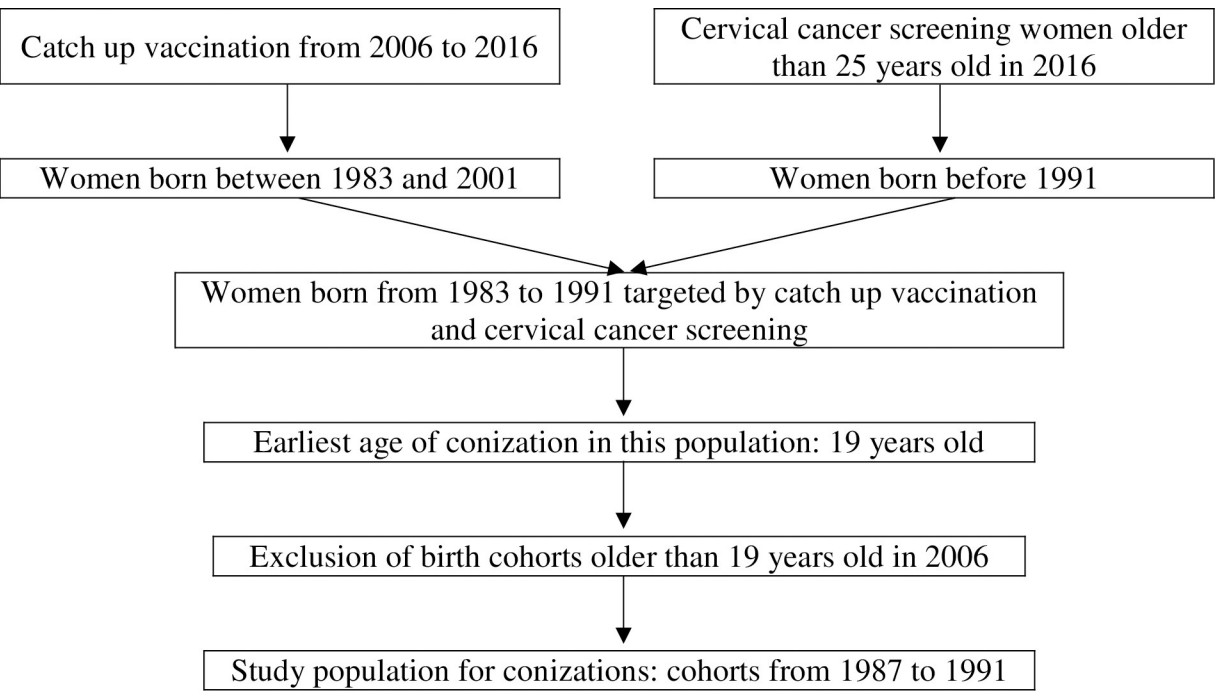

**Fig 2. Chart flow summarizing the selection of population.**

Chi-square tests were used to compare categorical variables, t-tests to compare continuous variables and the Logrank test was used to test the probability of conization over time between the vaccinated and non-vaccinated populations. All data analyses were performed using R software (R Foundation for Statistical Computing, Vienna, Austria, 2018, 3.5.1, https://www.R-project.org, Version 1.1.463 – © 2009–2018 RStudio, Inc).

## Results

The population included in the EGB over the period 2006 to 2016, including women born between 1983 and 1991, constituted a cohort of 42,452 women. Catch-up HPV vaccination coverage was 9.8% for at least one dose of vaccine (from 0% for the 1983 birth cohort to 30.9% for the 1991 cohort) and 5.8% for 3 doses of vaccine (0% to 20.6%) (Fig 3). Catch-up vaccination coverage therefore increased over time.

The non-vaccinated population was 38,323 women, and the vaccinated population was 4,129 women (Table 2). The mean age at the end of the study was lower in the vaccinated population (27.5 years) than in the non-vaccinated population (30.3 years, $p < 10^{-5}$). The percentage of patients treated for a chronic long-term illness (affection de longue durée: ALD) was not significantly different between the two groups: 5.7% in the overall population, 5.8% in the vaccinated population and 5.7% in the non-vaccinated population (p = 0.72). In France, long-term illness (ALD) refers to one of the thirty chronic diseases (hypertension, multiple sclerosis, etc.) on the list drawn up by the French health insurance system. It gives rise to full reimbursement of care related to this pathology.

During the follow-up period, 16 conizations were performed between the ages of 19 and 30 years in the vaccinated group versus 174 conizations in the non-vaccinated group. The conization rate between the ages of 19 and 30 years was significantly lower in the vaccinated group, with a Hazard Ratio of 0.59 (95% CI[0.39–0.90]; p = 0.043) (Fig 4A). The conization rate for

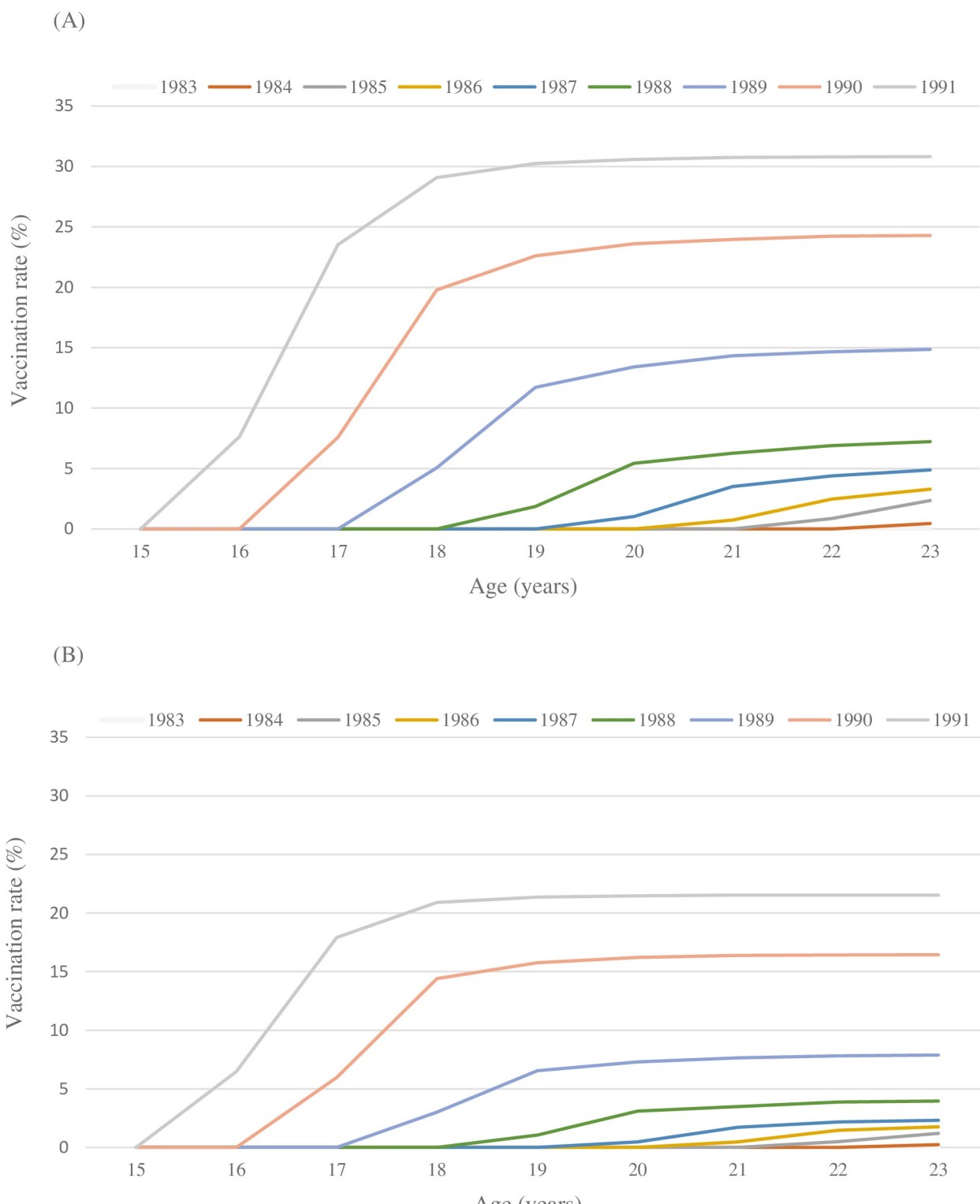

**Fig 3. Vaccination rate according to birth cohort and age.** (A) Cumulative HPV vaccine coverage for at least 1 dose. (B) Cumulative HPV vaccine coverage for 3 doses.

**Table 2. Characteristics of the overall population and the vaccinated and non-vaccinated subgroups.**

| | Overall (n = 42,452) | Vaccinated (n = 4,129) | Non-vaccinated (n = 38,323) | P |
|---|---|---|---|---|
| Mean age at the end of follow-up (years) | 30.0 | 27.5 | 30.3 | $<10^{-5*}$ |
| ALD° rate, n (%) | 2,408 (5.7%) | 239 (5.8%) | 2,169 (5.7%) | $0.72^{\alpha}$ |
| Vaccination rate for ≥ 1 dose, n (%) | 4,129 (9.7%) | | | |
| Vaccination rate for ≥ 3 doses, n (%) | 2,585 (6.1%) | | | |

Values are expressed as mean or number of patients (%).

°ALD: chronic long-term illness.

*t-test.

$^{\alpha}$Chi-square test.

patients over the age of 25 (birth cohorts from 1987 to 1991) was also calculated, corresponding to the population screened by Pap smears. In this population, the Hazard Ratio for conization rate was 0.57 (95% CI[0.32–1.01]; p = 0.052) (Fig 4B). At the age of 30 years, corresponding to the end of follow-up, conization had been performed in 0.18% of women in the vaccinated group versus 0.68% of women in the non-vaccinated group.

Health care utilization data are presented in Table 3. Patients in the vaccinated group had consulted their general practitioner more often (mean of 3.8 consultations per year before the age of 25 years compared to 2.5 for non-vaccinated patients, $p<10^{-3}$), but had consulted their gynecologist less often (mean of 0.60 consultations per year before the age of 25 years compared to 0.19 for vaccinated patients, $p<10^{-3}$). Vaccinated patients were younger at the age of the first Pap smear (21 years versus 24 years for non-vaccinated women, $p<10^{-3}$), and a higher proportion of women had their first Pap smear before the age of 25 years in the non-vaccinated group (13.5% versus 5.4%, $p<10^{-3}$).

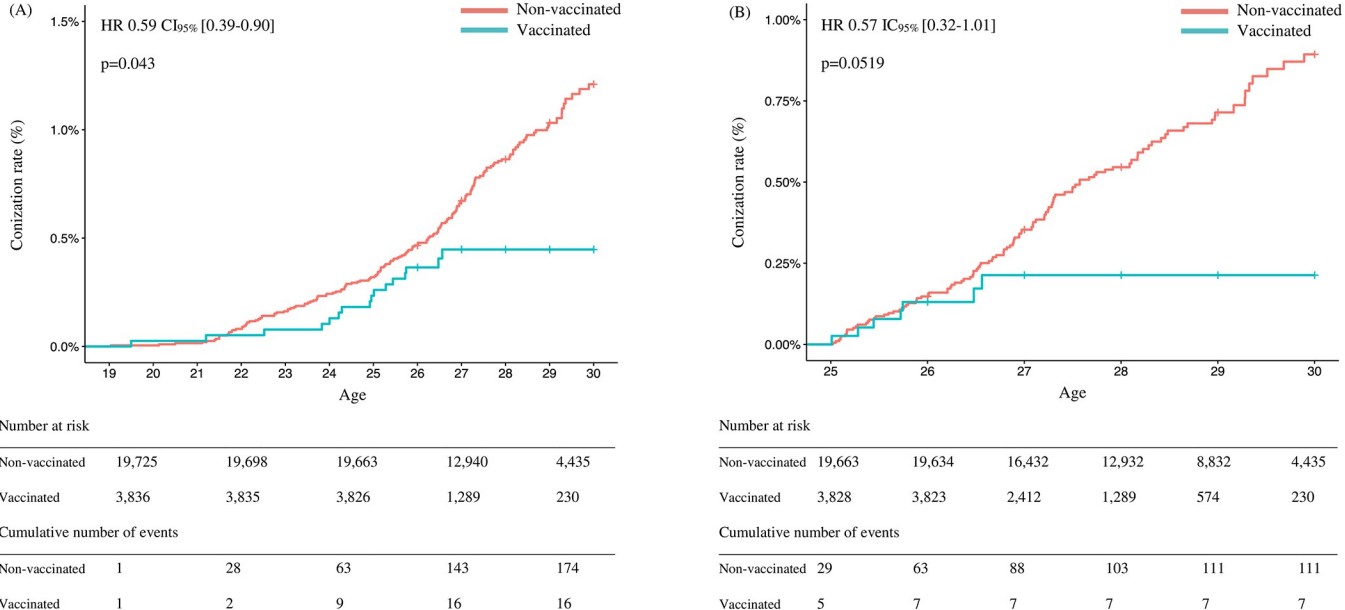

**Fig 4.** a. Conizations between the ages of 19 and 30 years for women born between 1987 and 1991. b. Conizations between the ages of 25 and 30 years for women born between 1987 and 1991.

**Table 3. Cares received by overall population and subgroups vaccinated and non-vaccinated.**

|  | Overall (n = 42,452) | Vaccinated (n = 4,129) | Non-vaccinated (n = 38,323) | P* |
|---|---|---|---|---|
| Annual number of family medicine consultations before the age of 25 | 2.7 (+/-0.003) | 3.8 (+/-0.001) | 2.5 (+/-0.003) | $<10^{-5}$ |
| Annual number of gynecologist consultations before the age of 25 | 0.6 (+/-0.001) | 0.2 (+/-0.002) | 0.6 (+/-0.001) | $<10^{-5}$ |
| Pap smear rate before the age of 25, n (%) | 5,378 (12.7%) | 223 (5.4%) | 5,155 (13.5%) | $<10^{-5}$ |
| Pap smear rate at any age, n (%) | 13,455 (31.7%) | 1,272 (30.8%) | 12,183 (31.8%) | 0.20 |
| Mean age at first Pap smear | 23.77 (+/-3.6) | 21.02 (+/-3.2) | 24.05 (+/-3.5) | $<10^{-5}$ |
| Mean age at conization | 26.90 (+/-2.9) | 25.29 (+/-2.8) | 26.98 (+/-2.9) | 0.005 |

Values are expressed as mean (+/- standard deviation) or number of patients (%).

*t-tests.

## Discussion

We report the first study on the impact of catch-up HPV vaccination in a real-life population in France, based on National Health Insurance database. In this cohort, consisting of 42,452 women, the average vaccination rate was 9.8%, lower than that reported in other countries for the same period [28–30].

The primary outcome, the conization rate, was significantly lower with an almost 50% reduction in the vaccinated group taking into account all conizations performed between the ages of 19 and 30 years.

The PATRICIA trial allowed approval of Cervarix® in various countries [31, 32]. This randomized trial involved 18,644 women aged 15 to 25 years, and studied the result of 3 doses of Cervarix® versus placebo. In the 4-year analysis, the conization rate was decreased by 25% (p = 0.0035). With a mean follow-up of 6.4 years, vaccination was 72% effective to prevent CIN2+ [33].

FUTUR I and II trials tested Gardasil® [34, 35]. FUTUR I included 5,455 women aged 16 to 24 years and FUTUR II included 12,167 women aged 15 to 26 years. Patients were randomized to a 3-dose vaccination arm or a placebo arm. In FUTUR I, no significant difference in the CIN2+ rate was observed after a follow-up of 3 years. In FUTUR II, 3 years after the first injection, intention-to-treat analysis showed a non-significant 17% reduction of CIN2+, while per protocol analysis demonstrated vaccination to be almost 100% effective.

The rates of cervical lesions were higher in these studies with a shorter follow-up than in our study. There are probably two reasons for this difference: the method of monitoring patients and the primary outcome. In the FUTUR trials, patients were followed by Pap smears starting 3 months after their inclusion in the study, may resulting in overdiagnosis [36].

The first Australian population study [15] included 14,085 non-vaccinated women and 24,871 women vaccinated with Gardasil® from two national registries. This study included women between the ages of 17 and 22 years. The risk of high-grade cytological lesions was lower in the vaccinated group with a HR of 0.72 (95% CI[0.58–0.91]). The second Australian study was a case-control study based on registry data [16]. It included 108,353 women aged 14 to 30 years at the time of their first Pap smear. The Gardasil® vaccine was administered as a 3-dose regimen. The odds ratio for exposure to 3 doses of vaccines was 0.54 (95% CI[0.43–0.67]) for CIN2+.

A Swedish study by Herweijer et al. also evaluated the effect of Gardasil® in a registry population [18]. This study included 1,333,691 women with a vaccination rate of 17.7% and 22,616 cases of CIN2+. Incidence rate ratios for CIN2+ were 0.25 when vaccination was performed before the age of 16 years and 0.78 when vaccination was performed after the age of 20 years, in favor of early vaccination.

The effectiveness of Gardasil[®] on dysplasia was studied using Danish registry data [19]. This study cohort consisted of 399,244 women, 62% of whom were vaccinated. There were 708 cases of CIN2+ across the entire cohort. A significant reduction of CIN2+ was observed in vaccinated women: HR was 0.56 and 0.27 (p = 0.005) for 1991–1992 and 1993–1994 birth cohorts, respectively. Either no events or no significant difference was observed in the other cohorts.

The results of our study are comparable to those of per protocol analysis of the initial studies and population-based studies conducted in other countries. This decrease in the conization rate in a catch-up population, which is not the target population of HPV vaccination, is encouraging, but also highlights the fact that the highest efficacy of vaccination is observed in the target population—i.e. before HPV exposure. In our study and in the PATRICIA study, the conization rate was lower in vaccinated patients, but this difference was only observed 2 or more years after vaccination, which could possibly be explained by the fact that vaccination does not prevent dysplasia in previously infected patients [37]. In the light of these data and the higher vaccination coverage rate in the target population (compared to the catch-up population), vaccination of the target population before the onset of sexual activity would be more effective. This effect can also be measured by the effectiveness of vaccination by age at vaccination as previously described [18]. Unfortunately, it was not possible to perform this type of analysis due to the low number of events in the vaccinated group.

Our study compared vaccinated women (regardless of the number of doses) with unvaccinated women, to fit the intention-to-treat design of a real-life study. Other studies used the same endpoint [15, 17], whereas the Australian study [16] compared the efficacy of each number of doses and the Swedish study [18] used the full schedule. The results of these studies were all consistent even though full vaccination had better efficacy. In our study, we evaluated health care consumption to investigate the determinants of vaccination and to control for potential biases. The higher health care consumption in the vaccinated group is probably a follow-up bias, as these patients consulted more often and were therefore more likely to be offered vaccination. As Pap smears and consultations were more frequent in this group, leading to more frequent and earlier diagnosis of cervical dysplasia, reduction of the conization rate is an even more encouraging result. The results were not adjustable for these potential biases due to the low number of events in the vaccinated group.

This study presents certain limitations: a small part of the French population (<8%) is not covered by the EGB data, and health care data collection is extensive, but its exhaustiveness and accuracy depend on precise transcription by health care professionals, especially coding of diagnoses and procedures. Another endpoint had to be used to study high-grade cervical dysplasia because the health data collected in the EGB do not provide any information about this diagnosis. Finally, another limitation of this study concerns the absence of "vaccination performed" data. Consequently, patients were considered to be vaccinated on the basis of vaccine reimbursement, rather than the actual vaccination procedure. This same type of bias is also observed in other population-based cohort studies. Vaccination is not always systematically performed as part of a vaccination program and the doses administered may not always be reliably recorded. Organized cervical cancer screening is not available in all of France and, even when it is available, individual screening remains frequent, possibly leading to differences in follow-up according to socio-economic categories, which also constitutes a vaccination bias [15–18].

This study of the efficacy of vaccination in the catch-up population may lack power in this population with a higher rate of pre-vaccination HPV infection than younger women corresponding to the primary indication. This weakness is also observed in other population-based studies that did not provide any information about the HPV status of the women before vaccination [15–18].

The EGB is a powerful tool, as it constitutes a random sample of 1/97th of the insured French population and, although the study was retrospective, data were collected prospectively. The use of a general population cohort provides real-life results; women are not monitored according to their vaccination status or screening status, so there is no risk of overmanagement. This real-life study on the French population therefore presents a high external validity.

The positive results of this study concerning the efficacy of HPV vaccination in the French population encourage an improvement in vaccination coverage, which is low in our study, as in previous studies. The main obstacles are: vaccine inequalities that can be superimposed on social inequalities [38], fear of side effects, poorly trained and informed physicians facing patients' refusal [39].

The possible levers of a public health strategy are a vaccination campaign, in schools for example, training of doctors who could better inform patients of the benefits and risks of vaccines, and a public communication campaign.A subsequent study conducted according to the same methodology could provide long-term results on the prevention of dysplasia and cervical cancer in this catch-up population, but also in the target population. These girls have better immunization against HPV and the HPV infection rate is lower than in the catch-up population, which would suggest a higher efficacy on the conization rate. This study provides more arguments in favour of a higher HPV vaccination rate in France.

## Acknowledgments

The authors gratefully acknowledge French National Health Insurance (CNAMTS) for providing data.

## Author Contributions

**Conceptualization:** Roman Rouzier.

**Data curation:** Sophie Houzard.

**Formal analysis:** Antoine Eliès, Claire Bonneau.

**Methodology:** Antoine Eliès, Claire Bonneau, Delphine Héquet.

**Software:** Sophie Houzard.

**Supervision:** Claire Bonneau, Roman Rouzier, Delphine Héquet.

**Writing – original draft:** Antoine Eliès, Claire Bonneau.

**Writing – review & editing:** Antoine Eliès, Claire Bonneau, Delphine Héquet.

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
