## [Decision Letter · Decision Letter 0]

11 Jun 2021

PONE-D-21-11439

Impact of catch-up human papillomavirus vaccination on cervical conization rate in a real-life population in France

PLOS ONE

Dear Dr. Elies,

Thank you for submitting your manuscript to PLOS ONE. After careful consideration, we feel that it has merit but does not fully meet PLOS ONE’s publication criteria as it currently stands. Therefore, we invite you to submit a revised version of the manuscript that addresses the points raised during the review process.

Beyond the minor language corrections and improvements this would be a good opportunity to expand the discussion on the indicated points as well as the limitations of the study.

We look forward to receiving your revised manuscript.

Kind regards,

Ivan Sabol

Academic Editor

PLOS ONE

Journal Requirements:

3.  Thank you for stating in the text of your manuscript "Data were extracted from the French National Health Insurance database called Echantillon Généraliste des Bénéficiaires (EGB), which is a permanent sample of the population covered by national health insurance". Please clarify whether this data was anonymized before researchers accessed it. Please also add all of this information to your ethics statement in the online submission form."

4. Thank you for including your manuscript text: "An institutional committee validates each research project." Please amend your current ethics statement to state:

a) whether ethics committee approval was specifically granted for this study

b) if so, include the full name of the ethics committee/institutional review board(s) that approved your specific study.

Additional Editor Comments (if provided):

P5L100 space issue “(24) .”

P6 L110 decimal comma used in table 1

P7 L141 Fig 2 flowchart is unclear. Right side box 1 requires women to be older than 25 in 2007. Next box states those are women born before 1991. However, a woman born in 1990 would be only 17 in 2007 instead of 25 as required by preceding box

P7 L147 statistical test used for Table 2 (for example) is not listed. Please consider stating all p values without an exponent and list the test used at least once for each stretch of p values shown.

P8 L159. Fig 3 lacks legend at least for panels/parts of the figure

P9 L171 Table 2 uses a thousand separator for large numbers, Tables 1 and 3 dont. Standard deviation is mentioned in the footer but not shown in Table 2.

P9 L187 Fig 4 uses decimal points and decimal comma inconsistently

P10L200 Table 3 it should probably be “mean age AT Pap/conization” instead of mean age of conization. In the header, non-vaccinated group is listed as “No-vaccinated”

P11 L208 inconsistent use of brackets for references “[26–28].”

P12 L246 it should probably be highlighted that most studies by other authors discussed in the discussion focused on 3 vaccine doses, while the current study mostly highlighted 1+ vaccine dose results

Reviewers' comments:

Reviewer's Responses to Questions

**Comments to the Author**

1. Is the manuscript technically sound, and do the data support the conclusions?

Reviewer #1: Partly

Reviewer #2: Yes

2. Has the statistical analysis been performed appropriately and rigorously? 

Reviewer #1: Yes

Reviewer #2: I Don't Know

3. Have the authors made all data underlying the findings in their manuscript fully available?

Reviewer #1: Yes

Reviewer #2: Yes

4. Is the manuscript presented in an intelligible fashion and written in standard English?

Reviewer #1: Yes

Reviewer #2: Yes

5. Review Comments to the Author

Reviewer #1: Re. the manuscript: Impact of catch-up human papillomavirus vaccination on 2 cervical conization rate in a real-life population in France

This is a retrospective real-life cohort study based on data collected by the French National Health Insurance. The aim of the study is to evaluate the impact of catch-up HPV vaccination on the rates of conization in France in a large population-based study.

page 3, line 39: It is stated that “Conization is part of the treatment of cervical dysplasia and cancer” … should probably be “Conization is part of the treatment of cervical dysplasia and early stage cancer

page 3, line 49: “group immunity” should rather be “herd protection” or “group pretection”

page 3, line 55-57: Relevant to mention that recently published papers on real world effectiveness against cervical cancer from Sweden and Denmark.

page 7, line 131: it is stated “The main outcome was a reduction of the conization rate between the 132 non-vaccinated and vaccinated populations” should probably be

“The main outcome was a comparison of the conization rate between the 132 non-vaccinated and vaccinated populations”

Nice study, however, based on very few exposed cases (N=16). The limitations should also include the fact that it was not possible to adjust for differences between vaccinated and non-vaccinated. In addition, it was not possible to consider age at vaccination, which has turned out to be an important factor when looking at effectiveness.

Have the authors tried to restrict the analysis even further to birth cohorts with higher coverage – e.g. birth cohorts 1989-1991? or will you then loose too much power. The women belonging to birth cohorts with a low coverage may have specific reasons to be vaccinated – in other words, they may have a different behavior/risk factor profile linked to a higher or lower exposure to HPV pre-vaccination

Reviewer #2: An interesting article which describes a potential clinical impact of HPV vaccine in France through the observation of conisation data derived from a national source

The article is well written and in general the data support the conclusions drawn

There are some issues with the manuscript, however, that would benefit from address:

The authors state on p5 that Gardasil AND Cervarix were the vaccines offered over the period of the evaluation, but I could not see any information as to what the estimated relevant proportions of the separate vaccines were. This would be valuable data as there is some evidence to indicate differential performance of the vaccines particularly in relation to cervical disease outcomes

The authors do not present information on deprivation - was a proxy of this not available from the national data set?

I understand that it is not a key deliverable of the manuscript but some more discussion of why vaccine uptake rates are low in France compared to other countries would be welcome and also implications for herd immunity (or lack thereof)

There are a couple of unusual phrases - eg page 12, line 250 "HPV contamination" arguably should be "HPV exposure" - I am also not sure what a "General scheme population" is, as described on page 13

6. PLOS authors have the option to publish the peer review history of their article (what does this mean?). If published, this will include your full peer review and any attached files.

Reviewer #1: No

Reviewer #2: **Yes: **Kate Cuschieri

---

## [Author Response · Author response to Decision Letter 0]

18 Nov 2021

Editor Comments:

3. Thank you for stating in the text of your manuscript "Data were extracted from the French National Health Insurance database called Echantillon Généraliste des Bénéficiaires (EGB), which is a permanent sample of the population covered by national health insurance". Please clarify whether this data was anonymized before researchers accessed it. Please also add all of this information to your ethics statement in the online submission form."

EGB data is pseudonymized before researchers access it. There is one identifier per patient, but no identifying data.

4. Thank you for including your manuscript text: "An institutional committee validates each research project." Please amend your current ethics statement to state:

a) whether ethics committee approval was specifically granted for this study

b) if so, include the full name of the ethics committee/institutional review board(s) that approved your specific study.

The Institut Curie, as a cancer center, has access to EGB data. Access is however restricted to researchers from the Institute who have validated training provided by the Health Insurance. Each project with analysis of these data is reviewed by an internal committee of Ethics and Science in Institut Curie. 

This study involved third-party data (National Health Insurance – EGB). Therefore, we do not have the rights to share these third-party data, in accordance with regulations SNDS (Système National des Données de Santé).

Law governing the dissemination of SNDS data : L. 1461-1 III du code de la santé publique R. 1461-1 du code de la santé publique

For more information : https://www.snds.gouv.fr/SNDS/Finalites-autorisees

Therefore we are in the “a)” situation, with legal restictions on sharing a de-identified data set.

To access the EGB data, please visit this website to apply for dataset : https://documentation-snds.health-data-hub.fr/introduction/03-acces-snds.html#les-acces-permanents

Additional Editor Comments (if provided):

P5L100 space issue “(24) .”

Corrected

P6 L110 decimal comma used in table 1

Corrected

P7 L141 Fig 2 flowchart is unclear. Right side box 1 requires women to be older than 25 in 2007. Next box states those are women born before 1991. However, a woman born in 1990 would be only 17 in 2007 instead of 25 as required by preceding box

This is a typo, in the box to the right must be at least 25 years old in 2016 to be eligible for screening 

P7 L147 statistical test used for Table 2 (for example) is not listed. Please consider stating all p values without an exponent and list the test used at least once for each stretch of p values shown.

All statistical tests were added in methods and in each table.

The R software cannot calculate a p-value lower than 2.2*10-16, as this value would not be very readable in a table we have replaced <2.2*10-16 by the value <10-5, which has a largely sufficient statistical value

P8 L159. Fig 3 lacks legend at least for panels/parts of the figure

Corrected

P9 L171 Table 2 uses a thousand separator for large numbers, Tables 1 and 3 dont. Standard deviation is mentioned in the footer but not shown in Table 2.

Corrected

P9 L187 Fig 4 uses decimal points and decimal comma inconsistently

Corrected

P10L200 Table 3 it should probably be “mean age AT Pap/conization” instead of mean age of conization. In the header, non-vaccinated group is listed as “No-vaccinated”

Corrected

P11 L208 inconsistent use of brackets for references “[26–28].”

Corrected

P12 L246 it should probably be highlighted that most studies by other authors discussed in the discussion focused on 3 vaccine doses, while the current study mostly highlighted 1+ vaccine dose results

A paragraph was added on this point to discuss the effect according to the number of doses administered P14 L263

Reviewers' comments:

Reviewer #1: 

page 3, line 39: It is stated that “Conization is part of the treatment of cervical dysplasia and cancer” … should probably be “Conization is part of the treatment of cervical dysplasia and early stage cancer

The sentence has been corrected

page 3, line 49: “group immunity” should rather be “herd protection” or “group pretection”

Replaced by herd immunity

page 3, line 55-57: Relevant to mention that recently published papers on real world effectiveness against cervical cancer from Sweden and Denmark.

The reference has been added with a comment P3 L57

page 7, line 131: it is stated “The main outcome was a reduction of the conization rate between the 132 non-vaccinated and vaccinated populations” should probably be

“The main outcome was a comparison of the conization rate between the 132 non-vaccinated and vaccinated populations”

The sentence has been corrected

Nice study, however, based on very few exposed cases (N=16). The limitations should also include the fact that it was not possible to adjust for differences between vaccinated and non-vaccinated.

In addition, it was not possible to consider age at vaccination, which has turned out to be an important factor when looking at effectiveness.

The manuscript has been revised to discuss these two points P14 L273 and 259 

Have the authors tried to restrict the analysis even further to birth cohorts with higher coverage – e.g. birth cohorts 1989-1991? or will you then loose too much power.

The exclusion of cohorts poses several problems: decrease of power, the study is no longer addressing all catch-up vaccination but just a part, increase of family-wise error rate.

 The women belonging to birth cohorts with a low coverage may have specific reasons to be vaccinated – in other words, they may have a different behavior/risk factor profile linked to a higher or lower exposure to HPV pre-vaccination

The increase of the vaccine coverage in time is explained by the fact that there is no vaccine campaign but a vaccine by individual approach (coming from the doctor or the patient), therefore the implementation of the vaccination is slow. This low vaccine coverage is addressed in the introduction, we have developed this "French" problem in the discussion P15 L302 with references added

Reviewer #2: 

The authors state on p5 that Gardasil AND Cervarix were the vaccines offered over the period of the evaluation, but I could not see any information as to what the estimated relevant proportions of the separate vaccines were. This would be valuable data as there is some evidence to indicate differential performance of the vaccines particularly in relation to cervical disease outcomes

Unfortunately, the only information available in the data extracted from this reimbursement database was the number of dose and the dates of dispensing. The type of vaccine was not included in the data.

The authors do not present information on deprivation - was a proxy of this not available from the national data set?

The EGB is a health database that does not contain socio-economic data

I understand that it is not a key deliverable of the manuscript but some more discussion of why vaccine uptake rates are low in France compared to other countries would be welcome and also implications for herd immunity (or lack thereof)

There is no vaccine campaign in France but a vaccine by individual approach (coming from the doctor or the patient). This low vaccine coverage is addressed in the introduction, we have developed this "French" problem in the discussion P15 L302

There are a couple of unusual phrases - eg page 12, line 250 "HPV contamination" arguably should be "HPV exposure" - I am also not sure what a "General scheme population" is, as described on page 13

The word has been replaced. The notion of general regime was clarified in the discussion: it corresponds to more than 92% of the population, i.e. less than 8% excluded

---

## [Decision Letter · Decision Letter 1]

4 Feb 2022

PONE-D-21-11439R1Impact of catch-up human papillomavirus vaccination on cervical conization rate in a real-life population in FrancePLOS ONE

Dear Dr. Elies,

Thank you for submitting your manuscript to PLOS ONE. After careful consideration, the reviewers brought up a minor issue in the discussion that should be addressed before acceptance. Therefore, we invite you to submit a revised version of the manuscript that likely will not undergo another round with the reviewers.

We look forward to receiving your revised manuscript.

Kind regards,

Ivan Sabol

Academic Editor

PLOS ONE

Journal Requirements:

Reviewers' comments:

Reviewer's Responses to Questions

**Comments to the Author**

1. If the authors have adequately addressed your comments raised in a previous round of review and you feel that this manuscript is now acceptable for publication, you may indicate that here to bypass the “Comments to the Author” section, enter your conflict of interest statement in the “Confidential to Editor” section, and submit your "Accept" recommendation.

Reviewer #1: All comments have been addressed

Reviewer #2: All comments have been addressed

2. Is the manuscript technically sound, and do the data support the conclusions?

Reviewer #1: Yes

Reviewer #2: Yes

3. Has the statistical analysis been performed appropriately and rigorously? 

Reviewer #1: Yes

Reviewer #2: Yes

4. Have the authors made all data underlying the findings in their manuscript fully available?

Reviewer #1: Yes

Reviewer #2: No

5. Is the manuscript presented in an intelligible fashion and written in standard English?

Reviewer #1: Yes

Reviewer #2: Yes

6. Review Comments to the Author

Reviewer #1: The authors have adequately responded to the comments. However, the new sentence on p. 3:

"More recently, a large Swedish cohort study showed a reduced risk of invasive

cervical cancer among girls and women vaccinated (20)" should be something like ...

So far, only two large cohort studies studies - one from Swedish (20) and one from Denmark (ref) have shown a reduced risk of invasive cervical cancer among girls and women vaccinated"

Ref: Real-World Effectiveness of Human Papillomavirus Vaccination Against Cervical Cancer.

Kjaer SK, Dehlendorff C, Belmonte F, Baandrup L.

J Natl Cancer Inst. 2021 Oct 1;113(10):1329-1335

As cervical cancer is the ultimate end point, it is important to mention both of the only two papers which have currently documented effectiveness in this regard.

Reviewer #2: The manuscript is improved as a result of the revision(s) - i also accept the limitations the authors explain in relation to dissemination of the original data

7. PLOS authors have the option to publish the peer review history of their article (what does this mean?). If published, this will include your full peer review and any attached files.

Reviewer #1: No

Reviewer #2: No

---

## [Author Response · Author response to Decision Letter 1]

10 Feb 2022

Response to Reviewers

4. Have the authors made all data underlying the findings in their manuscript fully available?

Reviewer #1: Yes

Reviewer #2: No

Response :

The reviewer did not specify in his comments why he disagrees on this point. All necessary information has been provided in the texts and forms regarding restrictions on publicly sharing data 

6. Review Comments to the Author

Reviewer #1: The authors have adequately responded to the comments. However, the new sentence on p. 3:

"More recently, a large Swedish cohort study showed a reduced risk of invasive

cervical cancer among girls and women vaccinated (20)" should be something like ...

So far, only two large cohort studies studies - one from Swedish (20) and one from Denmark (ref) have shown a reduced risk of invasive cervical cancer among girls and women vaccinated"

Ref: Real-World Effectiveness of Human Papillomavirus Vaccination Against Cervical Cancer.

Kjaer SK, Dehlendorff C, Belmonte F, Baandrup L.

J Natl Cancer Inst. 2021 Oct 1;113(10):1329-1335

As cervical cancer is the ultimate end point, it is important to mention both of the only two papers which have currently documented effectiveness in this regard.

Response :

The Danish study has been added to the manuscript, thank you for this comment.

Reviewer #2: The manuscript is improved as a result of the revision(s) - i also accept the limitations the authors explain in relation to dissemination of the original data

Thank you.

---

## [Editor Report · Decision Letter 2]

18 Feb 2022

Impact of catch-up human papillomavirus vaccination on cervical conization rate in a real-life population in France

PONE-D-21-11439R2

Dear Dr. Elies,

We’re pleased to inform you that your manuscript has been judged scientifically suitable for publication and will be formally accepted for publication once it meets all outstanding technical requirements.

Kind regards,

Ivan Sabol

Academic Editor

PLOS ONE
---

## [Editor Report · Acceptance letter]

2 Mar 2022

PONE-D-21-11439R2 

Impact of catch-up human papillomavirus vaccination on cervical conization rate in a real-life population in France 

Dear Dr. Eliès:

I'm pleased to inform you that your manuscript has been deemed suitable for publication in PLOS ONE. Congratulations! Your manuscript is now with our production department. 

Kind regards, 

on behalf of

Dr. Ivan Sabol 

Academic Editor

PLOS ONE